



# Relations between cyclones and ozone changes in the Arctic using data from satellite instruments and the MOSAiC ship campaign

Falco Monsees[1], Alexei Rozanov[1], John P. Burrows[1], Mark Weber[1], Annette Rinke[2], Ralf Jaiser[2], and Peter von der Gathen[2]

[1]Institute of Environmental Physics (IUP), University of Bremen, Bremen, Germany
[2]Alfred Wegener Institute, Helmholtz Centre for Polar and Marine Research, Potsdam, Germany

**Correspondence:** Falco Monsees (monsees@iup.physik.uni-bremen.de)

**Abstract.** Large-scale meteorologic events (e.g. cyclones), referred to as synoptic events, strongly influence weather predictability but still cannot be fully characterised in the Arctic region because of the sparse coverage of measurements. Due to the fact that atmospheric dynamics in the lower stratosphere and troposphere influence the ozone field, an approach to analyse these events further is the use of space-borne measurements of ozone vertical distributions and total columns. In this

study we investigate the link between cyclones and changes in stratospheric ozone by using a combination of unique measurements during the MOSAiC ship expedition, ozone profile and total column observations by satellite instruments (OMPS-LP, TROPOMI), and ERA5 reanalysis data. Three special cases during the MOSAiC expedition were selected and classified for the analysis. They consist of one 'normal' cyclone, where a low surface pressure coincides with a minimum in tropopause height, and two 'untypical' cyclones, where this is not observed. The influence of cyclone events on ozone in the upper-troposphere

lower-stratosphere (UTLS) region was investigated, using the fact that both are correlated with tropopause height changes. The negative correlation between tropopause height from ERA5 and ozone columns was investigated in the Arctic region for the three-month period from June to August 2020. This was done using total ozone columns and subcolumns from TROPOMI, OMPS-LP and MOSAiC ozonesonde data. The greatest influence of tropopause height changes on ozone contour levels occurs at an altitude between 10 and 20 km. Moreover, the lowering of the 250 ppb ozonopause (about 11 km altitude) below 9 km

was used to identify and track cyclones using OMPS-LP ozone observations. The potential of this approach was demonstrated in two case studies where the boundaries of cyclones could be determined using ozone observations. The results of this study can help improve our understanding of the relationship between cyclones, tropopause height, and ozone in the Arctic and demonstrate the usability of satellite ozone data for investigating cyclones in the Arctic.

## 1 Introduction

The Arctic is a region that is particularly sensitive to climate change and has experienced dramatic changes in recent decades (Stroeve and Notz, 2018; Box et al., 2019). More frequent and intense synoptic events, such as cyclones, are one effect of climate change in the Arctic (e.g. Rinke et al., 2017; Day et al., 2018; Zhang et al., 2023). These events not only affect local weather conditions but generate preferred patterns of seasonal circulation in the Arctic and can contribute to monthly-to-



seasonal large-scale circulation patterns (Wernli and Papritz, 2018; Graversen and Burtu, 2016). Measuring and monitoring
synoptic events in the Arctic is therefore of great importance to gain a better understanding of the role of the Arctic in the
global climate. Sea level pressure, air temperature, wind speed and direction, and total column water vapour are primarily used
as atmospheric parameters to characterise such events (Rinke et al., 2021). Another important parameter is ozone, which is a
dynamic tracer of troposphere-stratosphere interactions. Since cyclones exert a large influence in this altitude region, ozone is
affected by them (Millán and Manney, 2017). Because the Arctic is a remote and difficult-to-access region, it is challenging to
make accurate and comprehensive measurements. Existing observing stations are limited and the coverage of measurements is
often inadequate (Curry et al., 2004). Ozone satellite data provide an alternative way to observe changes in the Arctic induced
by synoptic events because it offers high coverage and continuous observations that goes beyond the limited measurements
available on land and at sea (Veefkind et al., 2012; Flynn et al., 2014).

Synoptic event-induced ozone changes were first investigated by Dobson et al. (1929), who studied the connection between
weather events and their influence on the total ozone column. Not long after, Meethan and Dobson (1937) showed a significant
correlation between the total ozone column and the tropopause height. This is explained by the fact that a downward movement
of the air masses causes an increase in the total ozone column because the ozone is sucked into the column, and an upward
movement causes a reduction in the total ozone column because the ozone is pushed out of the column (Reed, 1950; Barsby
and Diab, 1995; Zou and Wu, 2005). This negative correlation between tropopause height and ozone columns (e.g. Appenzeller
et al., 2000; Steinbrecht et al., 1998, 2001) enables us to investigate weather phenomena that influences atmospheric dynamics
using ozone data (Zou and Wu, 2005). For example, Orsolini et al. (1998) identified storm tracks during winter and spring
by analysing daily gridded Total Ozone Mapping Spectrometer (TOMS) ozone data. Davis et al. (1999) developed a method
for determining three-dimensional winds from TOMS ozone data. Using high-resolution mesoscale numerical modelling and
Total Ozone Mapping Spectrometer-Earth Probe (TOMS-EP) observations, Olsen et al. (2000) diagnosed a strong cyclone in
the midwestern United States. They also found that the total ozone column distribution closely resembled the geopotential
height at the 350-hPa surface. Furthermore, Jang et al. (2003) used TOMS ozone data to predict cyclones and found that
assimilating TOMS measurements into a mesoscale model had a positive influence on the prediction of an east coast winter
storm that occurred in January 2000 (Zou and Wu, 2005).

In this study, we investigate to what extent satellite-based ozone data can contribute to the analysis of cyclones in the Arctic.
To this end we use a combination of data from satellite instruments (TROPOMI, OMPS-LP), the Fifth Generation of ECMWF
Atmospheric Reanalysis (ERA5) (Hersbach et al., 2020) and the Multidisciplinary drifting Observatory for the Study of Arctic
Climate (MOSAiC) ship campaign (Shupe et al., 2020). In particular, we use ozone data from the Ozone Mapping and Profiler
Suite Limb Profiler (OMPS-LP) and the Tropospheric Monitoring Instrument (TROPOMI), due to a good vertical resolution
of the former and a high horizontal resolution and dense sampling of the latter data set. The combination of space-borne
observations and in-situ measurements from the research vessel (RV) *Polarstern* makes it possible to obtain a comprehensive
picture of the ozone distribution in the Arctic during cyclones.

The paper is structured as follows. First an overview of the data from the satellite instruments, ERA5 reanalysis and the
MOSAiC ship campaign (Section 2) along with the methods to determine the tropopause is presented (Section 3). Then, we



investigate three selected cyclone events, which were identified and classified during the MOSAiC campaign using ERA5 data,
to verify the relationship between cyclones, tropopause shift, and ozone changes in Section 4.1. Subsequently, the OMPS-LP
and TROPOMI satellite data are used to check the consistency of the ERA5 data and to verify the general relationship between
cyclones, tropopause shift, and ozone changes in the Arctic(Section 4.2). Finally, the potential and limitations of using satellite
ozone data to assess cyclones are examined using selected case studies (Section 4.3).

## 2 Data

### 2.1 ERA5 reanalysis

ERA5 assimilates measurements of different atmospheric variables (e.g. wind, temperature, surface pressure, ozone, water
vapour) from ground stations and satellites into a numerical weather prediction model. Ozone profile and column data from the
satellite instruments GOME, GOME-2, MIPAS, MLS, OMI, SBUV, SBUV-2, SCIAMACHY, and TOMS are input for ozone
assimilation (Hersbach et al., 2020). In addition, brightness temperatures of AIRS, CRIS, HIRS, and IASI are assimilated,
which depend on ozone. It covers the period from 1979 to the present and is continuously updated. ERA5 is described in
detail in Hersbach et al. (2020) and offers improved performance over the Arctic compared to other reanalyses (Graham et al.,
2019a, b). For this study, we use the high-resolution data ($0.25° \times 0.25°$) at a time resolution of one hour. Total ozone column,
ozone profile, surface pressure, pressure levels, temperature, air density and potential vorticity were used. Potential vorticity
and temperature were used to determine the dynamic and thermal tropopause height using methods described below. The
conversion of the vertical coordinate from pressure to altitude was done by first calculating the geopotential with a combination
of the hydrostatic equation and the ideal gas law, to then calculate the altitude from the geopotential with the use of the gravity
constant.

### 2.2 Ozonesonde data from the MOSAiC ship campaign

MOSAiC took place from September 2019 to October 2020. The German RV *Polarstern* was used as a platform for a multidis-
ciplinary research and during the expedition comprehensive data were collected (Shupe et al., 2022). As part of it, ozonesondes
were regularly launched from the *Polarstern* to measure the vertical distribution of ozone in the atmosphere (von der Gathen
and Maturilli, 2020a, b). These sondes have a vertical resolution of 3 to 10 meters, providing a very reliable source of ozone
data up to an altitude of typically 25 to 30 km.

### 2.3 Satellite ozone data

OMPS-LP is an instrument aboard NASA's Suomi National Polar-orbiting Partnership (Suomi-NPP) satellite, which measures
solar light scattered in the Earth's atmosphere in the UV-visible-NIR spectral range (Flynn et al., 2014). Suomi-NPP's orbit
is sun-synchronous, passing the equator at 13:30 local time. Due to its limb viewing geometry, where the scattered sunlight is
measured tangentially to the Earth's surface, OMPS-LP is capable of providing accurate measurements of ozone in the upper





troposphere and lower stratosphere (UTLS) region with a moderately high vertical resolution of about three km in the Arctic
UTLS region (Arosio et al., 2022).

TROPOMI on board the European Space Agency's (ESA) Sentinel-5 Precursor (S5P) satellite provides accurate measurements of column trace gas amounts (e.g. ozone, methane, formaldehyde) in the Earth's atmosphere. S5P follows NPP in the same orbit about 5 minutes apart. TROPOMI measures in the nadir viewing geometry, where the scattered sunlight is measured in the sub-satellite direction. It provides high horizontal resolution of 3.5 km × 7 km, which changed to 3.5 km × 5.5 km in
August 2019, and the across-track swath is 2600 km wide (Veefkind et al., 2012). The TROPOMI total column ozone product used here is WFDOAS V4 retrieved at our institute (Weber et al., 2022).

## 3  Methods

### 3.1  Tropopause

The tropopause separates the troposphere and the stratosphere, is usually characterised by an abrupt change in the vertical
temperature gradient and atmospheric composition, and has a large influence on ozone columns and stratospheric ozone (James, 1998; Millán and Manney, 2017). There are different definitions of the tropopause. The conventional definition is the thermal tropopause, which is defined by the World Meteorological Organization (WMO) as the lowest altitude level at which the temperature lapse rate decreases to 2 K/km or less and does not exceed this value within 2 km above this altitude (Hoinka, 1997; North et al., 2015).

There are several methods to determine the dynamical tropopause height. One common method is based on a potential vorticity (PV) threshold. The PV is a quasi-conservative quantity in the atmosphere that can be derived from the vertical and horizontal wind distribution, temperature, and atmospheric pressure. PV generally increases with altitude with a maximum gradient in the UTLS, which was originally defined as the dynamical tropopause (Reed, 1955; Kunz et al., 2011). Depending on latitude and season a PV threshold between 1 and 4 PVU (Potential Vorticity Units), is proposed instead to define the
dynamical tropopause (Hoerling et al., 1991; Xian and Homeyer, 2019). In this study, we employed the upper threshold of this range, 4 PVU, because it is less disturbed by small-scale disturbances evident at surfaces of lower PV values.

According to Chrgian (1967), the ozonopause is defined as the point at which the ozone content starts rapidly rising. It therefore separates the ozone-poor troposphere from the ozone-rich stratosphere (Lapeta et al., 2000). Like the dynamical tropopause, it is defined by a threshold value. According to Bethan et al. (1996) it is the lowest level, at which the ozone volume
mixing ratio (VMR) surpasses 80 parts per billion (ppb) with the additional criteria that immediately above the ozonopause the VMR is larger than 110 ppb and the average vertical gradient in the 200 m layer above is higher than 60 ppb/km. Layers of stratospheric air within the troposphere are excluded by these criteria. Ivanova (1972) discovered that the difference between the altitudes of the tropopause and ozonopause varies by approximately 1 km in 85% of cases, but there are instances where the difference can be as much as 4-9 km. Since the 80 ppb ozonopause is normally located below the thermal and dynamical
tropopause and is generally below the valid range of OMPS-LP ozone data, a higher threshold value (250 ppb) compared to 80 ppb is used for the evaluation of cyclones, as discussed later.





## 4 Results

### 4.1 Case study of three cyclone events using ERA5 data

After Rinke et al. (2021) identified and classified all cyclone events that impacted the *Polarstern* during the MOSAiC expe-
dition, three of them were selected for the initial evaluation of cyclones in the Arctic. Two of these events were selected for
their 'untypical' structure and one for comparison with a 'normal' structure. A normal structure is characterised by the center
of the cyclone coinciding with a low tropopause. This event (Event 1) took place between 15 and 17 of November 2019. The
two 'untypical' events, where this connection was not observed, occured between 30 January to 2 February 2020 (Event 2)
and between 15-17 April 2020 (Event 3). The aim of this initial evaluation was to gain a basic understanding of the influence
of cyclones on ozone in the UTLS region in the Arctic and to check whether cyclones with an untypical structure also show a
connection to ozone.

Figure 1 shows the time-altitude cross-section of the ozone VMR and the 4-PVU dynamical tropopause height, both from
ERA5 data, at the position of the *Polarstern* for the 'normal' event 1. In addition, the ERA5 surface pressure is shown as
an indicator of the cyclone. The expected coincidence between the centre of the low surface pressure area and the minimum
tropopause height can be clearly seen at 6 UTC on 17 November (marked as a dashed black line; Fig. 1). The tropopause lowers
from about 9 km to about 5 km with a time delay of about 6 hours from the minimum surface pressure. One can also see a
relationship between the tropopause height change and the vertical distribution of ozone. The ozone contour levels around the
tropopause and up to an altitude about 10 km descend. The ozone VMR level of 150 ppb, for example, moves down during the
cyclone from around 10 to 6 km. This can, as already explained earlier, be attributed to the fact that ozone rich air masses are
sucked into the column above the tropopause when the tropopause descends.



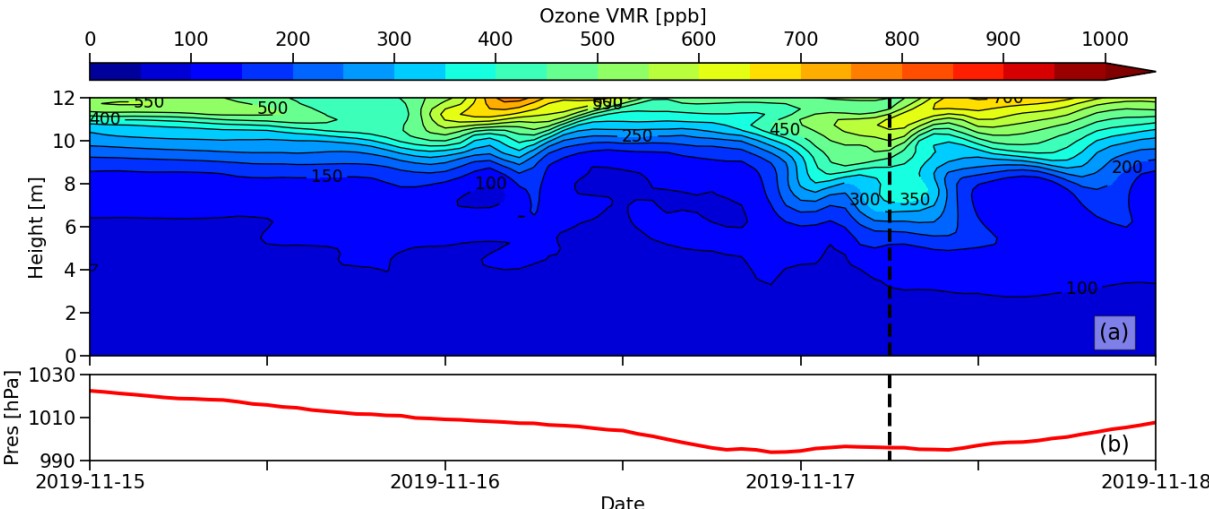

**Figure 1.** Upper panel (a): Time-height cross-section of ozone volume mixing ratio (ppb) and 4-PVU dynamical tropopause height (red curve) from ERA5. Lower panel (b): corresponding surface pressure from ERA5. The data are at the *Polarstern* location from 15 to 18 November 2019 (Event 1).

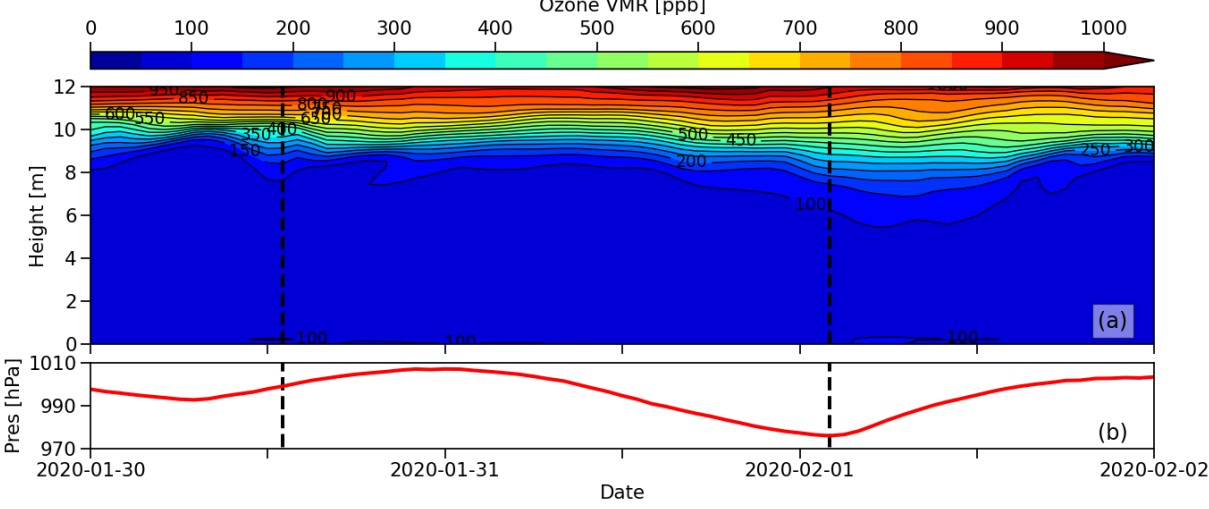

**Figure 2.** The same as Fig. 1 but for the period from 30 January to 2 February 2020 (Event 2).

For the 'untypical' Event 2, as illustrated in Figure 2, the correlation between ozone content and tropopause height is much less pronounced but still visible. For example, at 13 UTC on 30 January the tropopause lowers from around 10 to 8 km and the corresponding ozone VMR level 100 ppb from around 9 to 8 km. This lowering can possibly be attributed to a preceding other




cyclone, that impacted the *Polarstern* at 7 UTC on 30 January. In addition, there is no visible disturbance of the tropopause
near the centre of the second minimum of surface pressure at 2 UTC on 1 February (Fig. 2).

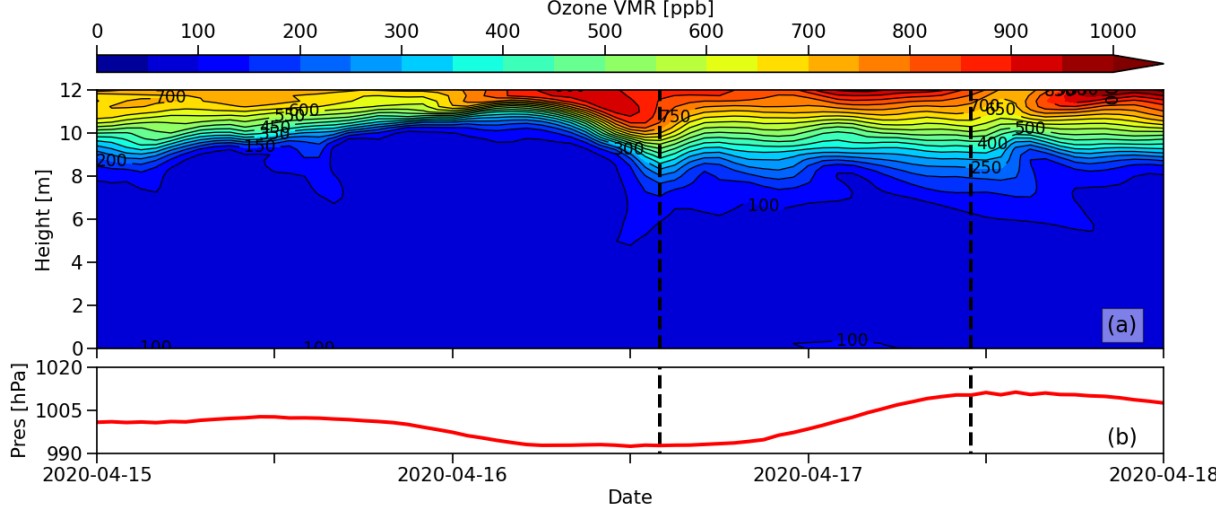

**Figure 3.** The same as Fig. 1 but for the period from 15 to 18 April 2020 (Event 3).

For the 'untypical' Event 3, however, the tropopause is lowered near the minimum of the surface pressure at 14 UTC 16
April (Fig. 3). The tropopause moves down from about 11 to 7.5 km. The correlation between the tropopause and the ozone
content still exists as the ozone VMR level of 150 ppb moves down from around 10 to 7.5 km. In contrast, a minimum in
tropopause height is seen later near a maximum in surface pressure around 12 UTC on 17 April, suggesting that the tropopause
disturbance is decoupled from the surface pressure.



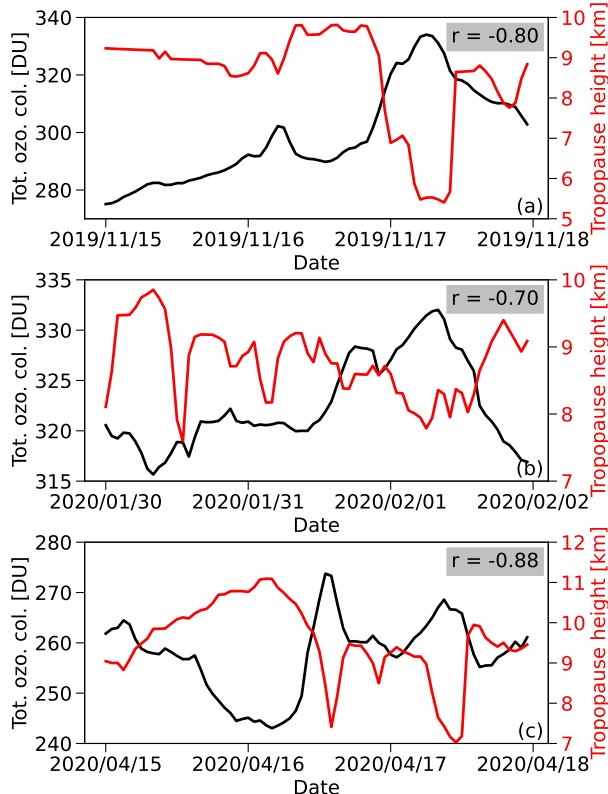

**Figure 4.** Total ozone column (black) and 4-PVU dynamical tropopause height (red) from ERA5 at the *Polarstern* position of (a) Event 1, (b) Event 2, and (c) Event 3.

Figure 4 shows the evolution of the total ozone column and the 4-PVU dynamical tropopause, both from ERA5, for the three events. For all events, the expected negative correlation between the total ozone column and the tropopause height can be seen. Consequently, for Event 1, we have a negative correlation of $r = -0.80$ and the highest ozone column at the location of the cyclone (Fig. 4a, November 17th, 6am). The total ozone column increases from 290 DU (November 16th, 12am) to 334 DU (November 17th, 6am) and the tropopause moves down from approximately 10 to 5.5 km during this event. Due to the smaller scale of the tropopause height, it can be seen that also for Event 2 (Fig. 4b), which is classified as 'untypical', a slight lowering of the tropopause by about 1 km occurs at 4 UTC on 1 February. At this point, the total ozone column is at its maximum, rising from approx. 325 DU (23 UTC on 31 January) to 332 DU (February 1st, 10am). The negative correlation is a slightly lower compared to Event 1 with a correlation coefficient of $r = -0.70$. For Event 3 (Fig. 4c) we have the highest absolute correlation of the three events ($r = -0.88$) and the highest total ozone column change, rising from approx. 243 DU (April 16th, 4am) to 274 DU, coinciding with a local minimum of the tropopause (14 UTC on 16 April).





## 4.2 Extended analysis of tropopause/ozone linkage during the MOSAiC summer

Due to the fact that tropopause lowering can be detected during normal and untypical cyclone events, although the strength of the lowering varies, we assume that most cyclone events can be identified from a change of tropopause height. Therefore, the

question arises to what extent satellite ozone data can be used for the detection of tropopause-induced ozone changes.

### 4.2.1 S5P/TROPOMI total ozone columns

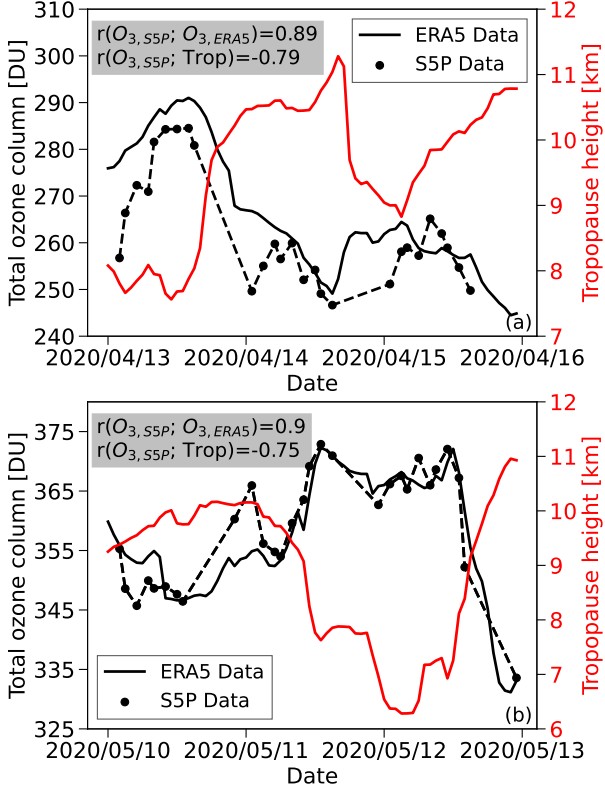

**Figure 5.** Total ozone column from ERA5 and S5P data and 4 PVU dynamical tropopause height from ERA5 at the *Polarstern* position for (a) a rising tropopause in April 2020 and (b) a descending tropopause in May 2020.

To start off with the analysis of the S5P satellite ozone data, two different cases of tropopause movement were used to consider both a tropopause rising and a tropopause lowering (Fig. 5). The rising tropopause (anticyclone) took place in April 2020, and the lowering tropopause (cyclone) took place in May 2020. When comparing the total ozone column from ERA5 and S5P,

we see a good qualitative agreement and a high correlation between both datasets (r=0.89 for the case in April and r=0.90 in May 2020, respectively). One can observe that for the anticyclone-induced tropopause rise (Fig. 5a), which occurred from 13 to 15 April 2020, an expected decrease in the S5P total ozone column was observed. The tropopause rises from about 8 to



11 km and the total ozone column from ERA5 and S5P drops from about 290 to 250 DU. The same correlation is shown for the cyclone-induced descending tropopause (Fig. 5b), which occurred from 11 to 13 May 2020, when we have an increase in

the S5P and ERA5 total ozone column. The tropopause sinks approximately from 10 km to 6 km and the total ozone column increases from about 350 DU to 370 DU. In addition, there is a high overall anti-correlation between the 4 PVU dynamical tropopause height from ERA5 and total ozone column from S5P for both 3-day periods (r=−0.78 in April and r=−0.76 in May).

To make a more robust statement about the correlation between S5P total ozone column and the ERA5 dynamical tropopause

height, a scatter plot for the three-month period June to August 2020 at the *Polarstern* position is shown in Figure 6a. This time period was chosen to make the evaluation comparable with the evaluation of the OMPS-LP data, which only has observations at the *Polarstern* position from June onwards due to the long polar nights, the smaller swath of the instrument, and the northern position of the *Polarstern*. For the same reason, data north of 81.3°N are coloured red and not included in the correlation analysis of tropopause height versus S5P ozone. Most of the red data points found in the 240 to 280 DU range deviate from

the correlation line and occur in August 2020. This is most likely due to the minimum of stratospheric dynamic activity at the end of summer (e.g. Weber et al., 2011). The satellite measurements and ERA5 data were required to be collocated within a range of ±111 km and ±15 min around the *Polarstern* positions. This enables us to compare the satellite results with MOSAiC ozonesonde data, as shown later (Section 4.2.3). The anti-correlation between the ERA5 tropopause height and S5P total ozone column still remains high with a correlation coefficient of r= −0.83. Comparing the total ozone column from ERA5

and S5P data (Fig. 6b), ERA5 tends to underestimate the observed total ozone column at higher values, but still a very good agreement between the two datasets at the *Polarstern* positions is evident. A high correlation coefficient of r= 0.96 underlines the suitability of gap-free ERA5 ozone data as a reliable alternative to satellite data for cyclone- and anticyclone-related ozone change studies.



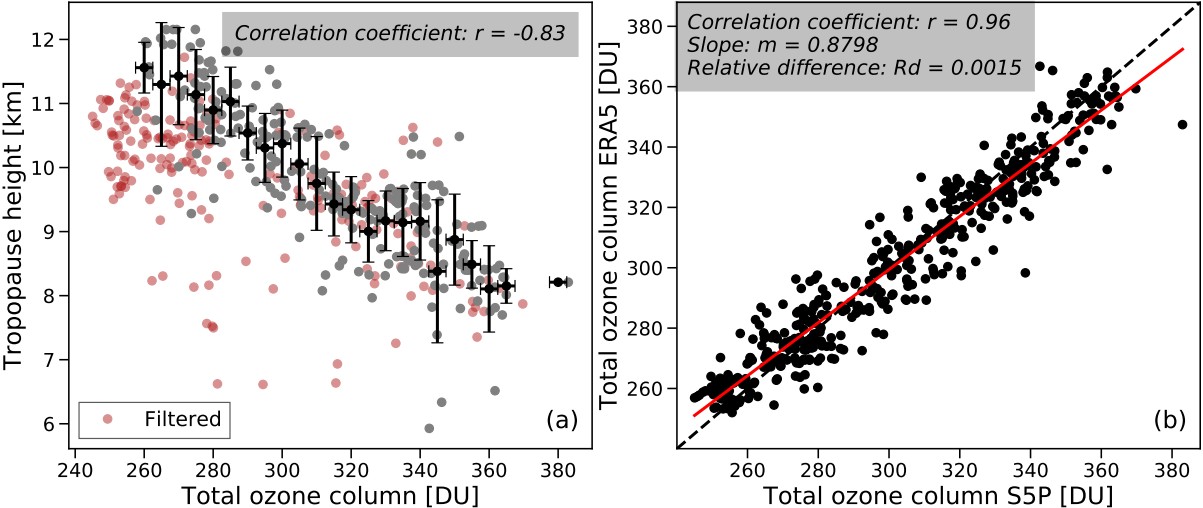

**Figure 6.** Scatter plots: (a) 4-PVU dynamical tropopause height from ERA5 vs. total ozone from S5P (grey and red data points). Data measured after 8 UTC on 15 August 2020 and data north of 81.3°N are coloured red, for which OMPS-LP data are not available. All other data points are shown in grey and used to determine the correlation coefficient. The black data points are averages in 5 DU bins of the grey data with vertical standard deviations as error bars; (b) Scatter plot of ERA5 and S5P total ozone columns. The data are at the *Polarstern* location during June-August 2020.

### 4.2.2 OMPS-LP ozone profiles

For this analysis, the time period of June to August 2020 was considered. As explained earlier, OMPS-LP ozone data are available only for latitudes below 81.3°N during this period. No OMPS-LP data collocated with the MOSAiC ship campaign were available after mid-August 2020. OMPS-LP measurements within a ±555 km range around the *Polarstern* and within a range of ±30 min around MOSAiC collocated ERA5 data were selected. Figure 7 shows the partial ozone column from 10 to 20 km from ERA5 and OMPS-LP along with the 4-PVU tropopause height. The partial ozone column from 10 to 20 km was

used here because the tropopause height change has the highest influence on ozone in this altitude region, as discussed in more detail below. OMPS-LP is in a good qualitative agreement with the ERA5 ozone and has a high correlation of r=0.87. Moreover, a high anti-correlation (r=−0.64) between the partial ozone column from OMPS-LP and ERA5 dynamical tropopause height is evident, especially on 24th June and 12th July 2020. During the June event, the tropopause height rises to 11.1 km and the partial ozone column from OMPS-LP falls to 80 DU. This effect is even stronger in the July event with a tropopause height of

12 km and a low partial ozone column of 66 DU.





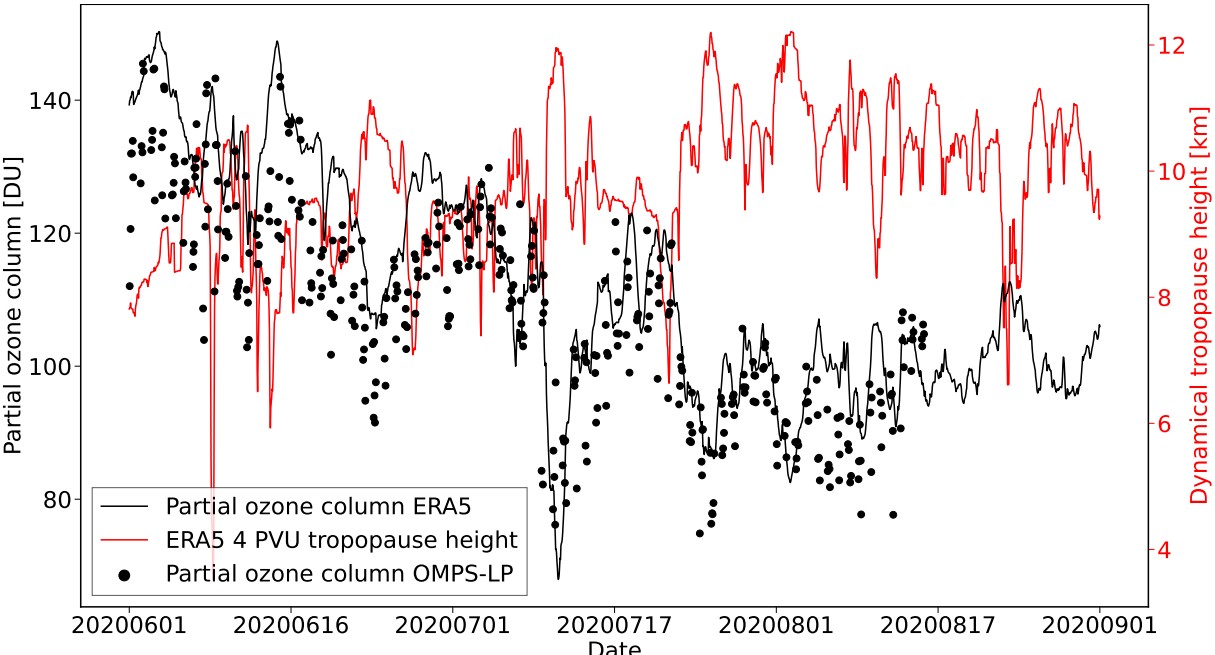

**Figure 7.** Partial ozone column in the altitude range 10-20 km from OMPS-LP and ERA5 and the ERA5 4-PVU dynamical tropopause height from June to August 2020 at the *Polarstern* position.

In order to find out which height region of ozone is most affected by tropopause height changes, partial ozone columns at different altitude ranges (10-20 km, 20-30 km, 30-60 km, and 10-60 km) are plotted versus the ERA5 4-PVU dynamical tropopause height in Figure 8. The highest correlation (r=−0.72) occurs for ozone columns in the 10-20 km range, which is expected because of the proximity to the tropopause. The influence of the tropopause height on the ozone contour levels decreases with increasing altitude (James et al., 1997). The correlation coefficient decreases to r=−0.44 in the 20-30 km height range and is even lower (r=−0.27) above in the 30-60 km height range. The correlation for the total ozone column, which is closely represented by the height range from 10-60 km (panel (d)), is nearly as high as the correlation in the 10-20 km range with a correlation coefficient of r=−0.69. This correlation is somewhat lower than for S5P columns due to the more relaxed collocation criteria used for OMPS-LP to obtain a sufficient number of collocated OMPS-LP observations. If the same collocation criteria is used for S5P with a limit of ±555km, a correlation coefficient of r=−0.53 is obtained.



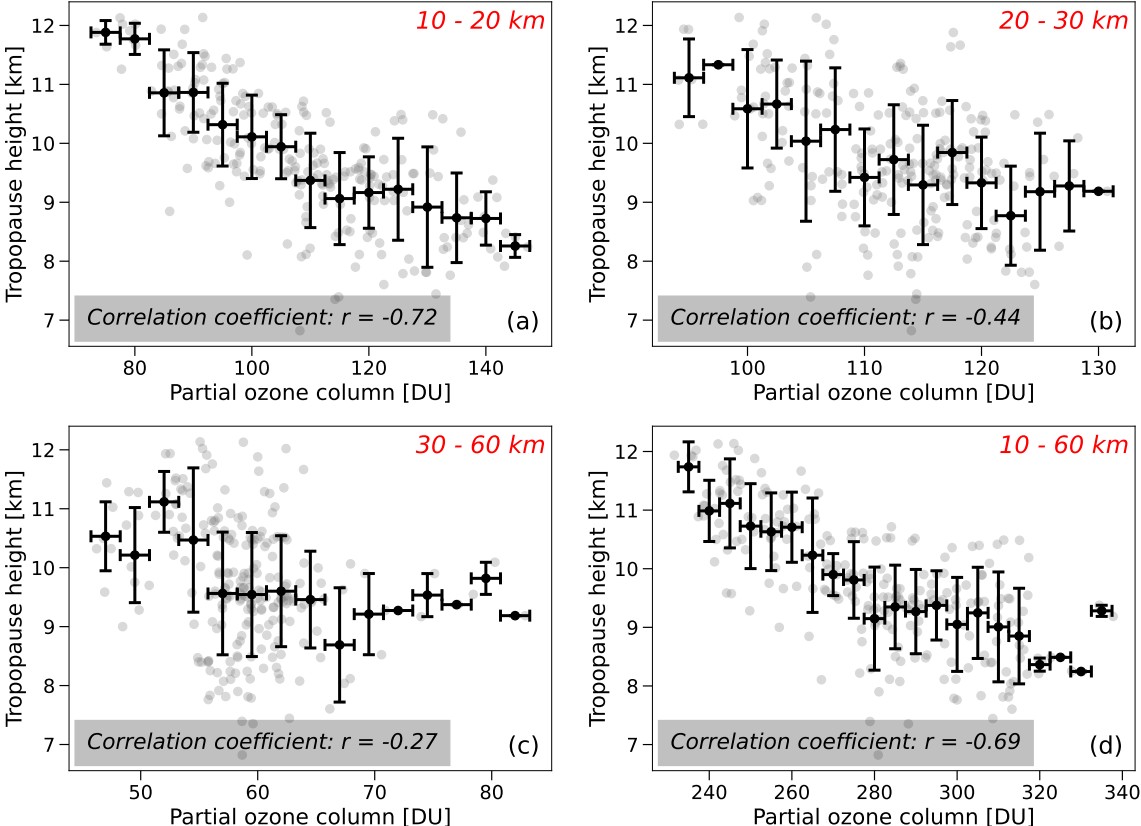

**Figure 8.** ERA5 4-PVU dynamical tropopause height versus OMPS-LP ozone subcolumns in the altitude ranges of 10-20 km (a), 20-30 km (b), 30-60 km (c), and 10-60 km (d) at the *Polarstern* location from June-August 2020 (grey data points). The black data points are averages in 5 DU bins of the grey data with vertical standard deviations as error bars. The grey data points were used to determine the correlation coefficients

### 4.2.3 MOSAiC ozone profile data

Figure 9a shows the correlation between the ERA5 4-PVU dynamical tropopause height and the ozone subcolumn (0 to 25 km) from MOSAiC ozonesondes for September 2019 to October 2020. There were in total 49 ozonesondes launched during the entire MOSAiC campaign that measured ozone at least up to an altitude of 25 km. The correlation between the tropopause

height and the ozone column is lower compared to OMPS-LP and S5P with a correlation coefficient of only r=$-0.51$, which is probably due to the fact that most of the ozonesonde launches were carried out on days when there were no cyclones above the *Polarstern* (Fig. 10). Comparing the integrated ozone columns from 0 to 25 km from ERA5 and MOSAiC (Fig. 9b), a high correlation (r=0.94) was found. The ERA5 data systematically overestimates the ozonesonde columns by about 1.9%.



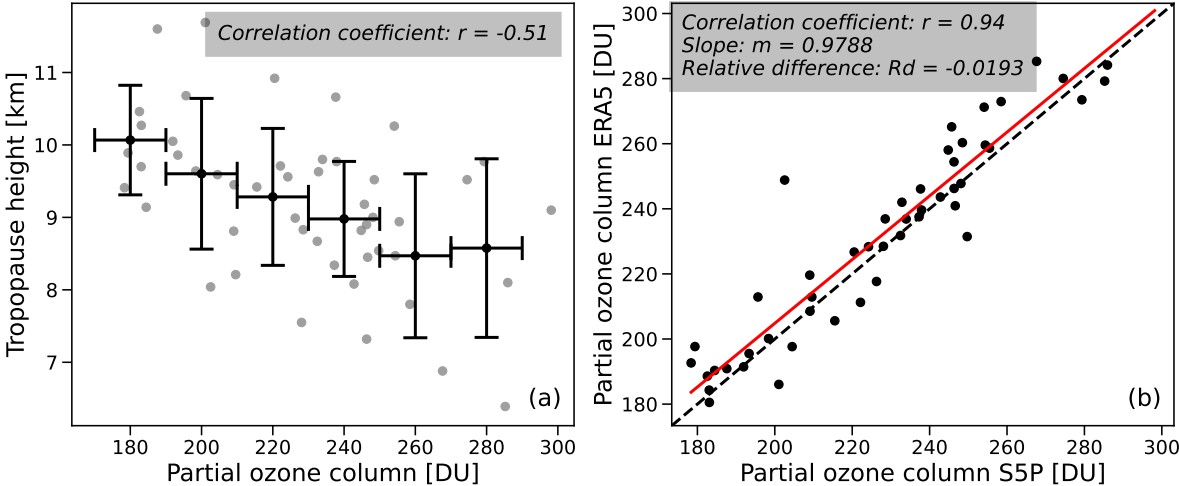

**Figure 9.** Scatter plots: (a) 4-PVU dynamical tropopause height from ERA5 versus the ozonesonde subcolumns (0-25 km) from MOSAiC. The black data points are averages in 20 DU bins of the grey data with vertical standard deviations as error bars. The correlation was determined from the grey points; (b) ERA5 versus MOSAiC ozonesonde subcolumns (0-25 km).

## 4.3 Cyclone identification with OMPS-LP ozone data

With the knowledge that most cyclone events are associated with a lowering of the tropopause and that the largest impact of tropopause height change on ozone is observed in the 10 to 20 km range, our next step was to find a way to use (stratospheric) ozone observations to identify cyclones. All cyclones that impacted the *Polarstern* during the MOSAiC campaign were classified in Rinke et al. (2021). An overview of the cyclones with their corresponding strengths is shown in Figure 10. The cyclone strength or depth is defined as the difference between the surface pressure at the center of the cyclone and the surface pressure

at the edge of the cyclone. The weakest and strongest cyclone had a depth of 1.57 and 44 hPa, respectively (Fig. 10). The cyclone from 7th to 13th May 2020 was not only the cyclone with the longest lifetime during the MOSAiC campaign but also the one with the highest cyclone depth of 44 hPa. Although there were no ozonesonde launches during this event, both ERA5 data and OMPS-LP ozone data were available to investigate this cyclone.



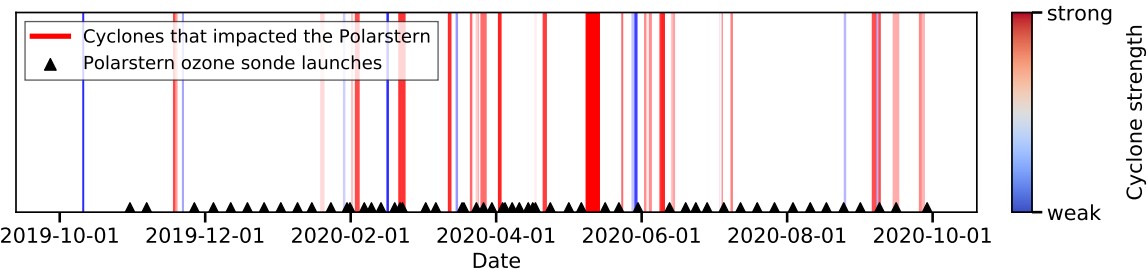

**Figure 10.** All cyclone events (vertical bars) as a function of time, that impacted the *Polarstern* and were classified in Rinke et al. (2021), are colour-coded according to the cyclone strength (see main text for definition). The bar widths represent the lifetime of the individual cyclones. The days where ozonesondes were launched from the *Polarstern* are marked by black triangles in the bottom.

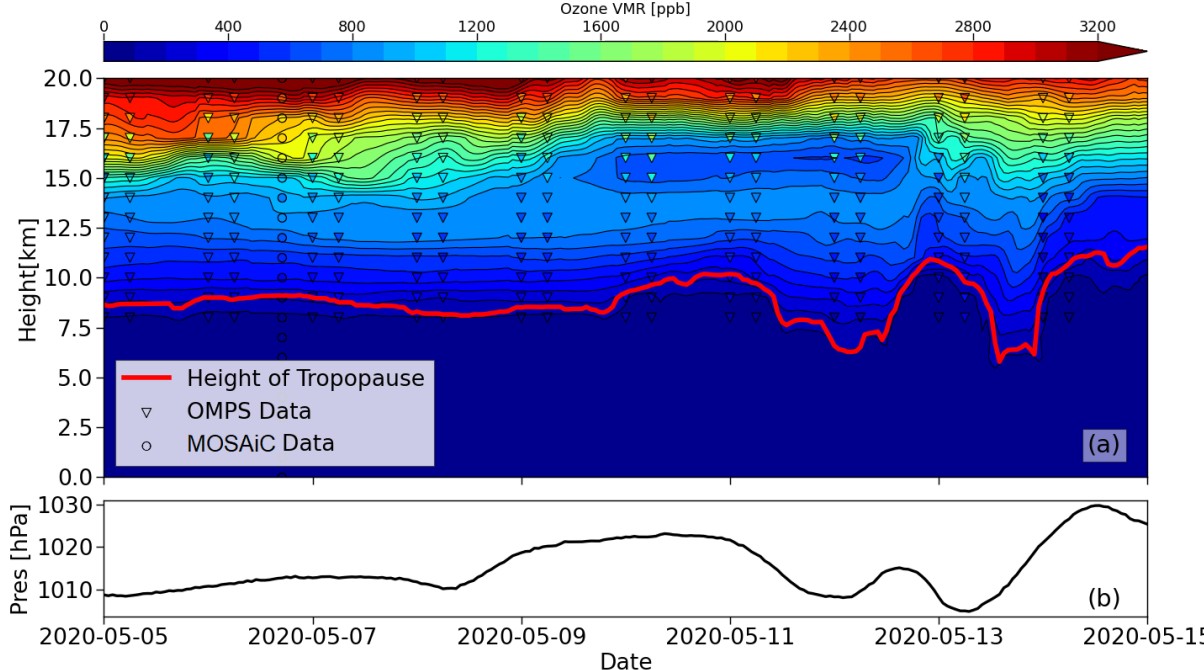

**Figure 11.** Panel (a): Time-height cross-section of ozone volume mixing ratio (ppb) from ERA5 (colour shading), OMPS-LP, and MOSAiC ozonesondes, and the dynamical 4-PVU tropopause height. The MOSAiC (circles) and OMPS-LP (triangles) ozone data have the same colouring as the contours of ERA5 data. Panel (b): Surface pressure from ERA5 at the *Polarstern* positions. Note the different colour range with respect to Figs. 1-3.

Figure 11 shows the time-height cross-section of ozone VMR from ERA5, OMPS-LP, and MOSAiC ozonesondes, the dynamical 4-PVU tropopause height, and the surface pressure from ERA5 data from 5-15 May 2020. Although the average



deviation between the OMPS-LP and ERA5 data is about 33% we still see an overall agreement between these two datasets in this time period. The data from the ozonesonde from *Polarstern* on 6th May agrees well with the ERA5 data with an average deviation of around 10%. The cyclone impacts the *Polarstern* on three occasions during this period: 8th, 12th, and 13th of May. The first occurrence is associated with only a small dip in the tropopause height and has no visible disturbance in the

ozone field. The second and third occurrences, on the other hand, are associated with a tropopause lowering of about 3-4 km associated with ozone contour levels descending in the altitude region between the tropopause and up to about 12 km for the second and 17 km for the third occurrence. On the third occurrence there is in addition a time delay of about 12 hours between the minimum of the surface pressure and the downward motion of the tropopause and ozone.

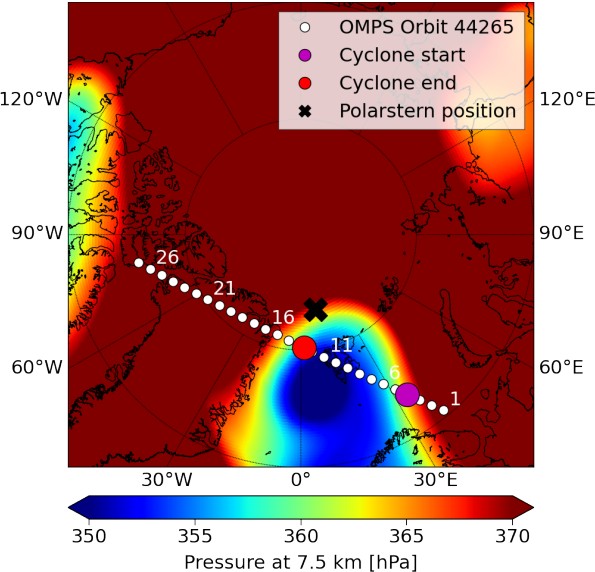

**Figure 12.** Pressure at 7.5 km from ERA5 in the Arctic at 9 UTC on 13 May 2020. The *Polarstern* position (black cross), the cyclone borders that were calculated using the ozonopause from OMPS-LP (big coloured dots) and OMPS-LP measurement points along orbit 44265 (small white dots) are shown.

To evaluate the influence of the third cyclone on ozone, one OMPS-LP orbit crossing the cyclone was selected (Fig. 12).

The start and end points of the cyclone crossing as determined from the ozonopause are also shown. The method for their determination is discussed below. Figure 13 shows the time-height cross-section of the ozone VMR from this OMPS-LP orbit (orbit number 44265) and the corresponding pressure at 7.5 km altitude from ERA5. Pressure at 7.5 km will be used to identify the cyclone instead of the surface pressure, on the one hand, because of the increasing surface altitude over Greenland, which strongly influences the surface pressure, and, on the other hand, because in this altitude range we observe the highest cyclone-

induced drop in pressure. As discussed before, ozone contour levels up to 15 km are influenced by the cyclone (Fig. 13). Due to the fact that OMPS-LP data are not available below 8 km altitude, a descending of ozone below this altitude is not visible in the figure.





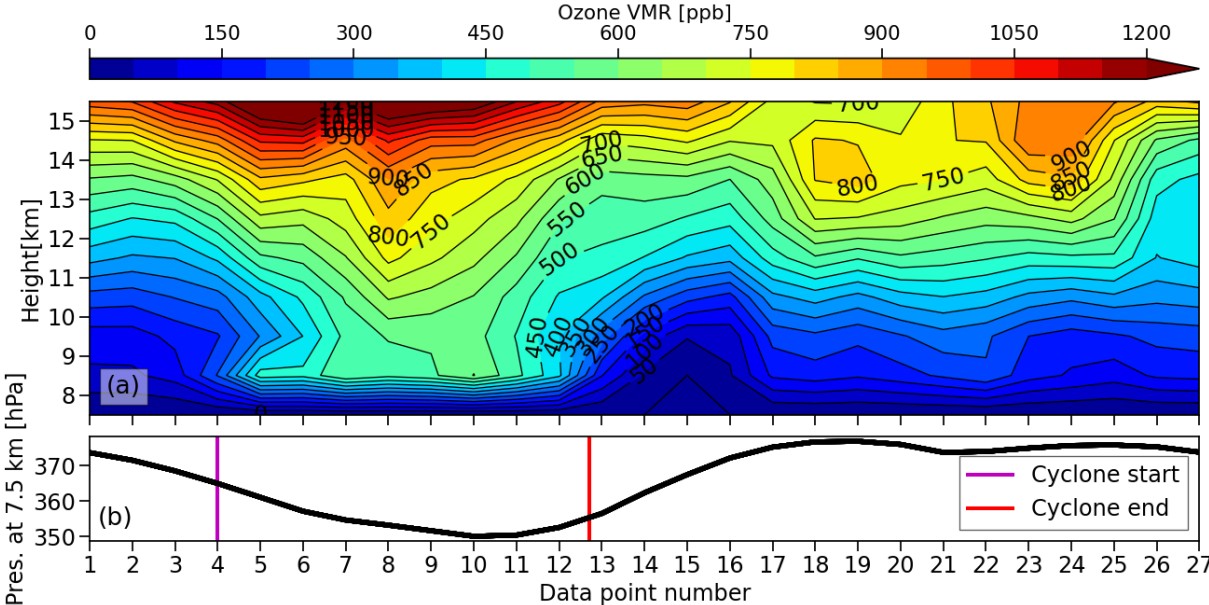

**Figure 13.** Panel (a): Time-height cross-section of ozone volume mixing ratio (ppb) from the OMPS-LP orbit 44265 at 9 UTC on 13 May 2020. Panel b: corresponding pressure at 7.5 km altitude with the cyclone borders derived from the ozonopause from OMPS-LP data (see main text). The x-axis is defined by the measurement point number of the partial OMPS-LP orbit.

Ozone contour levels at altitudes between 8 and 15 km are most strongly influenced by cyclones and their vertical movement can be tracked using OMPS-LP data. This can be exploited for an automated identification of cyclone boundaries by tracing
the movement of a certain ozone level. Different ozone VMR thresholds for defining the ozonopause were tested. To illustrate their differences, Figure 14a shows the time-height section of OMPS-LP ozone volume mixing ratio from orbit 44264, which is the preceding orbit to the one shown in Figure 13, on 13th May 2020 and selected ozone contour levels (80 ppb, 150 ppb, and 250 ppb) as candidates for the ozonopause. The panel b of Figure 14 again shows the pressure at 7.5 km with cyclone borders of the same cyclone event as illustrated in Figure 13. We notice that the 250 ppb contour best follows the cyclone-induced ozone
movement. The main reason is that the OMPS-LP data below 8 km are not retrieved, while ozone contours of 80 and 150 ppb can be located below that altitude. In this case we wouldn't see additional lowering of ozone contour levels, meaning that the 80 and 150 ppb ozone contour levels related to cyclones cannot be observed. The use of the 250 ppb contour as ozonopause is, therefore, more suitable as it is low enough in altitude to be impacted by most cyclones but high enough to have continuous reliable data from OMPS-LP.
The location when the 250 ppb ozone level falls below 9 km altitude was selected to define the extent of the cyclone. An altitude of 9 km was used because the 250 ppb contour normally lies between 10-12 km. To mitigate the influence of stratospheric streamers and other possible errors, we introduce the criterion that the ozone contour level 250 ppb has to be below 9 km for at least five consecutive measurements of OMPS-LP (approximately 1500 km along the orbit) in order to

N



qualify as a cyclone event. When applying this criterion to the OMPS-LP orbit 44265 (Figs. 12 and 13), we see that the ozone-
defined borders of the cyclone are reasonably defined when compared to the ERA5 pressure at 7.5 km. Although the end of
the cyclone seems to be somewhat too early, the main structure and strongest region of the cyclone is well captured using our
ozone criterion.

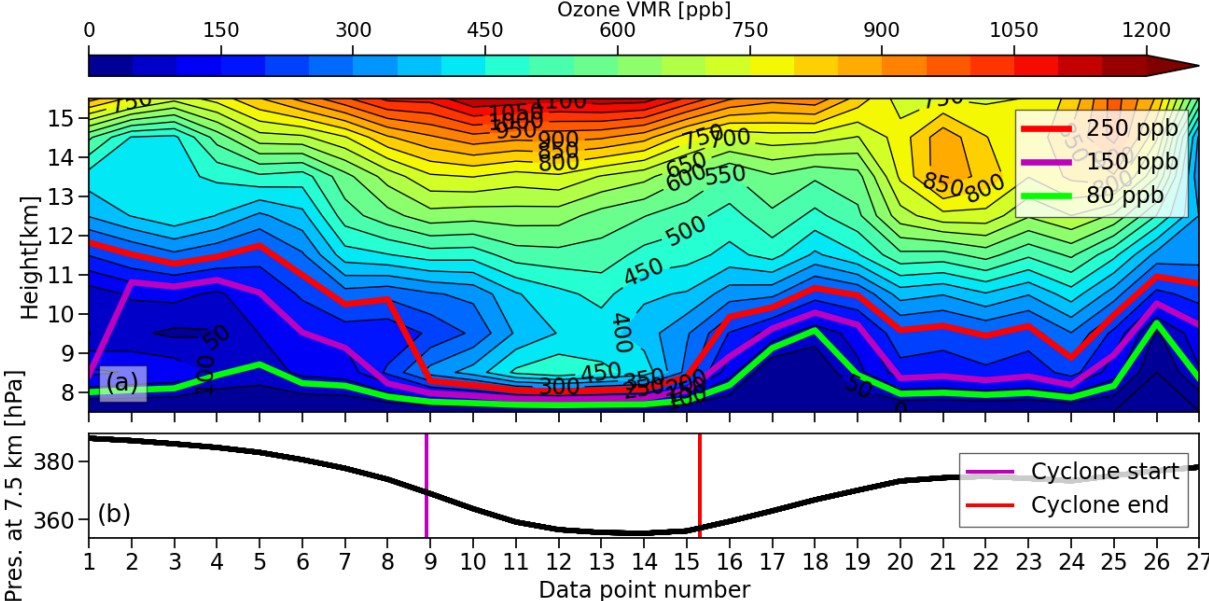

**Figure 14.** Panel a: Time-height section of ozone volume mixing ratio (ppb) from the OMPS-LP orbit 44264 at 7 UTC on 13 May 2020.
Different ozone levels (80, 150 and 250 ppb are shonw. Panel b: Pressure at 7.5 km with the cyclone borders that were calculated from the
250 ppb ozone level going below 9 km (see main text). The x-axis is defined by the measurement point number along the OMPS-LP orbit.

Another case study of a cyclone event on May 4th demonstrates the usefulness of our ozone criterion as illustrated in
Figure 15. This case was selected because there were three OMPS-LP orbits within 4 hours that crossed the cyclone. These
275 three consecutive OMPS-LP orbits were used to determine the cyclone borders. For all three orbits, there is good qualitative
agreement between the structure of the cyclone from ERA5 pressure data and the cyclone borders from the motion of the
OMPS-LP 250 ppb ozone level. This illustrates that the motion of a cyclone can be traced using OMPS-LP ozone data,
provided there are sufficient OMPS-LP orbits surrounding the cyclone.



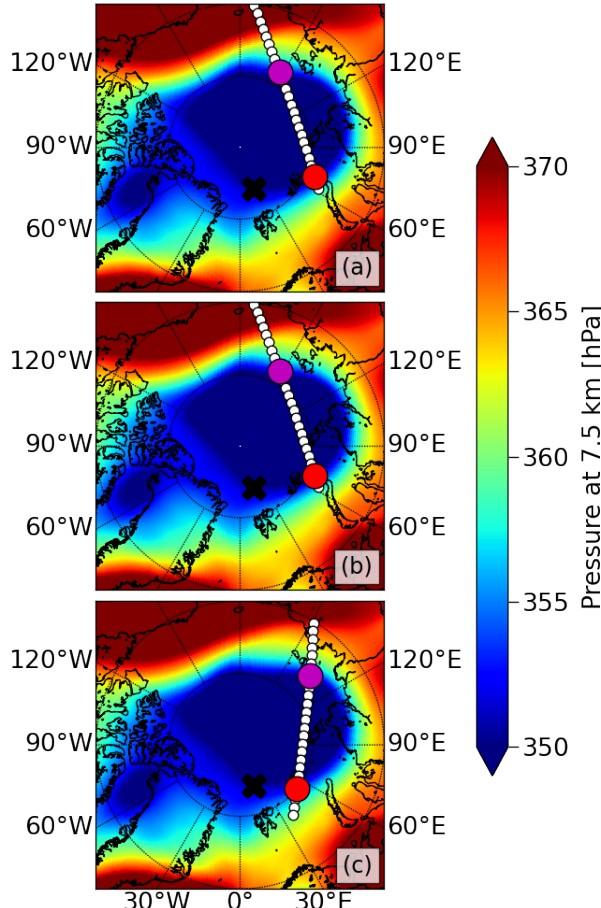

**Figure 15.** Pressure at 7.5 km for the cyclone event at (a) 0 UTC, (b) 2 UTC, and (c) 4 UTC on 4 May 2020. The position of the OMPS-LP measurement points of the corresponding OMPS-LP orbits and the calculated cyclone borders are marked (cf. Fig. 12).

## 5 Summary and conclusions

280 In this paper, we investigated the contribution of satellite-based ozone data for the analysis of cyclone events in the Arctic. We presented a method for determining cyclone borders using OMPS-LP ozone profile data. The criterion that the 250 ppb ozone level falls below 9 km at the cyclone boundary was found to be a reasonable choice. The potential of this method was demonstrated in two case studies during the MOSAiC campaign. The main rationale behind the ozone cyclone criterion is the fact that cyclone-induced changes in the tropopause height affect ozone above the tropopause by horizontal advection. The

285 connection between the total ozone columns and cyclone events was validated using the ERA5 4-PVU dynamical tropopause. The connection between tropopause changes and S5P total ozone column was not only confirmed for two case studies in April and May but also for a three-month period in the Arctic. This relationship was also investigated with MOSAiC and OMPS-



LP subcolumns. A lower correlation was found for the MOSAiC ozonesondes, due to the sparsity of data and the absence of ozonesonde launches during cyclones. Using the vertically resolved profiles from the OMPS-LP enabled us to study the influence of tropopause height changes on different altitude ranges. As expected the highest influence occurred at an altitude of 10 to 20 km (lowermost stratosphere). This influence was also confirmed with ERA5 pressure and ozone data for three cyclone events during the MOSAiC campaign. Contours of constant ozone VMR in the UTLS region follow the course of the descending tropopause during the cyclone events. Even in the case of untypical cyclone events (Rinke et al., 2021), a weak lowering of the tropopause was found above the cyclone. In one case, a time lag in subsidence of several hours after the surface pressure minimum during the MOSAiC campaign was noted, which requires further investigation. A significant correlation and good agreement between ozone data from OMPS-LP, S5P, MOSAiC ozonesondes and ERA5 ozone was shown, indicating that ERA5 ozone is suitable for investigating cyclone-induced ozone changes. ERA5 has the advantage that it is gap-free and covers a large period (at least since 1978) and there is no limitation due to the polar night like for OMPS-LP and S5P.

A method for determining cyclone borders was developed using OMPS-LP ozone. The best choice is the crossing of the 250 ppb levels to below 9 km altitude. This method was applied in two case studies using ERA5, MOSAiC ozonesondes, and OMPS-LP data. One event, according to Rinke et al. (2021) the strongest cyclone event during the MOSAiC campaign, was identified using OMPS-LP ozone observations (See Fig. 13). For the other one on May 4th, the cyclone border was properly identified in three consecutive OMPS-LP orbits (see Fig. 15).

Since this method has so far only been tested on cyclones, the question arises as to what extent the method is also suitable for other synoptic events (e.g. anticyclones), which needs further studies. It should be noted that this is only the first approach and the method may require additional analysis and improvements. For example, stratospheric streamers or other non-cyclone-based ozone motions may cause the 250 ppb level to drop below 9 km for a horizontal length of over 1500 km. In addition, the polar vortex have to be taken into account to reliably use ozonopause motion to evaluate cyclones.

*Acknowledgements.* This study was partly funded by the German Federal Ministry of Education and Research (BMBF) SynopSYS projects (FKZ 03F0872A and 03F0872B), the University and the State of Bremen. Large parts of the calculations reported here were performed at the HPC facilities of the Institute of Environmental Physics (IUP), University of Bremen, funded under the DFG/FUGG grants INST 144/379-1 and INST 144/493-1. The ozonesonde data reported in this manuscript were produced as part of the international Multidisciplinary drifting Observatory for the Study of Arctic Climate (MOSAiC) expedition with the tag MOSAiC20192020, with activities supported by *Polarstern* expedition AWI_PS122_00.The development of the satellite stratospheric ozone profiles by Carlo Arosio was supported by his ESA Living Planet Fellowship SOLVE and the PRIME program of the German Academic Exchange Service (DAAD) funded by the BMBF. We gratefully acknowledge the computing time the Resource Allocation Board granted and provided on the supercomputer Lise and Emmy at NHR@ZIB and NHR@Göttingen as part of the NHR infrastructure. The calculations for this research were conducted with computing resources under the project hbk00098.



*Data availability.* The L2 data set for OMPS-LP produced at the University of Bremen is available at the following link: https://doi.

org/10.5281/zenodo.7198052 (Arosio and Rozanov, 2022). Sentinel-5 Precursor TROPOMI data can be accessed through the Copernicus Open Access Hub at https://scihub.copernicus.eu. This dataset is openly available for public use, subject to the data policy. The MOSAiC ozonesonde data can be found at the following links: https://doi.org/10.1594/PANGAEA.919538 (Leg 1-3) (von der Gathen and Maturilli, 2020a) and https://doi.org/10.1594/PANGAEA.941294 (Leg 4-5) (von der Gathen and Maturilli, 2020a). The cyclone data are available at https://doi.org/10.1525/elementa.2021.00023 (Rinke et al., 2021). The ERA5 reanalysis data are available from the Copernicus Climate

Change (C3S) climate data store (CDS) at https://doi.org/10.24381/cds.adbb2d47 (Hersbach et al., 2020)

*Author contributions.* FM performed most of the data analysis and wrote the manuscript. ARo and MW supervised the study and contributed to writing the paper. JPB contributed to the review of the manuscript and the scientific outcome. ARi provided the dataset with all cyclone events that impacted the *Polarstern* and reviewed the paper. RJ, who leads the project, contributed to the review of the manuscript and the scientific outcome. PVDG provided the MOSAiC ozonesonde data and reviewed the paper. All the authors contributed to the discussion of

the paper and particularly the recommendations.

*Competing interests.* The contact author has declared that none of the authors have any competing interests.



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
