# Peer review of "Relations between cyclones and ozone changes in the Arctic using data from satellite instruments and the MOSAiC ship campaign"

_EGUsphere, 2023_

## Author Response (AR1)

**Relations between cyclones and ozone changes in the Arctic using data from satellite instruments and the MOSAiC ship campaign**

**F. Monsees[1], M. Weber[1], A. Rozanov[1], J. P. Burrows[1], A. Rinke[2], R. Jaiser[2,3], and P. von der Gathen[2]**

[1]Institute of Environmental Physics (IUP), University of Bremen, Bremen, Germany
[2]Alfred Wegener Institute Potsdam, Helmholtz Centre for Polar and Marine Research, Potsdam, Germany
[3]Department of Life, Light and Matter, University of Rostock, Rostock, Germany

monsees@iup.physik.uni-bremen.de

**Author's Response**

**We thank the reviewer for the helpful feedback on our manuscript! All comments were implemented in the paper. Our responses to the individual comments can be found below. The reviewer comments are given in red and our answers in black.**

**Referee #1**

This paper nicely shows that well known connections between ozone, potential vorticity, tropopause height, and, more generally, tropospheric weather also work in the Arctic. I think the paper is well written and well illustrated and deserves publication in ACP.

I have only few suggestions for changes.

1)      The color scale for ozone seems unfortunate in Figs. 1, 2, 3, 11. Especially in Figs. 1 to 3 everything is just dark blue and there is very little to see. Maybe a logarithmic color scale, say from 50 to 1000 ppb , would work much better. It would expand variations at low ppb, and compress variations at high ppb. I strongly suggest that the authors test this.

**Color scale and color scheme changed for Figs. 1,2,3,11 to display also the ozone distribution in the range 46 – 100 ppb.**

2)      Also in Figs. 1 to 3: I am missing/ not seeing the red line for the tropopause. Please add.

**Added the missing red line for the tropopause.**

3)      2.2 Vertical resolution of 3 to 10 meters. While ozone sondes may give data points every few meters, the time constant of the ozone reaction cell is about 20 seconds - which corresponds to a vertical resolution of about 100m for ozone. This is still much finer than OMPS Limb profiles, which also integrate over a few hundred kilometers horizontally. The finer structures seen by the sonde, and their intrinsic measurement noise may well explain another good part of the lower correlations mentioned later in 4.2.3.

**Changed the vertical resolution in the text to ~100m for the MOSAiC ozone sondes. Added a discussion (see section 3.2.3), that the finer structures could be another possible explanation for the lower correlation, but to test this we calculated the correlation between the ERA5 partial ozone column from 0 – 25 km and the ERA5 dynamical tropopause height for the same location and time as the MOSAiC ozone sondes and also got the same lower correlation of r = -0.51, which implies that the absence of cyclones are the main reason for the lower correlation.**

**Referee #2**

This manuscript outlined a method to identify cyclone passage using ozone and tropopause height measurements. Satellite, ERA5, and ozonesonde launch data during the MOSAiC expedition were used to both identify cyclones and evaluate the relationship between ozone levels, tropopause height, and cyclone passage. The authors found the 250ppb ozone level can best be used to identify cyclone passage.

This is an interesting additional way by which cyclones can be identified. The figures are clean and concise, and the writing is mostly clear with the exception noted below. There are places where clarification is needed, so the recommendation is major revisions. The following should be addressed/answered before publication.

1)      What value does this type of analysis bring in cyclone identification beyond using minima in sea level pressure/geopotential height or maxima in vorticity? I am not suggesting there is not value, just that it would strengthen the paper to highlight what value this brings beyond the cyclone tracking methods already available.

**The main goal of our analysis is to underline the statement, that we can get a dynamical information out of satellite ozone data that we can use for example for the assimilation in weather forecast models. The goal is not to use ozone data instead of pressure, wind speed, etc. for cyclone identification, but to show the usefulness of satellite ozone data as an additional parameter for the analysis of synoptic events. This has been made clearer throughout the paper. For this we have improved the Abstract, Sections 3.2.1 and 3.3, and the Summary & Conclusions by adding according additional sentences to better clarify the analysis goal.**

2)      A big question I had is – what is the large takeaway from this work? Verifying ERA5 ozone data so a gap-free, long term ozone dataset can be used for this type of analysis? A method for using ozone to identify cyclones? There was a lot of discontinuity in the description of the work, and how it all tied together. It would be helpful to be clearer as to why you did each step and better organize the analysis so that each step flows into the next. For example, there was the initial analysis for the 3 cyclones shown in figures 1-3, then different cyclones in Figures 11-15. There was the analysis looking at the relationship between tropopause height and total column ozone. And then the analysis comparing all of the ozone data to each other. There needs to be a bit more verbiage and reorganization to tie all of these analyses together. There are interesting results from this work - putting everything into context in a cohesive manner would really strengthen the paper.

A major takeaway from this work is to show the usefulness of satellite ozone data as an addition for the analysis of synoptic events. This was done by first showing that we have a connection between cyclones, tropopause height changes and ozone levels/total ozone columns also in the Arctic. We approach this first for the three cyclone cases (section 3.1), because two of them were untypical and we wanted to check the general validity of those connections. Indeed, we show that connections between cyclones, tropopause height change and ozone are found for cyclones of both typical and untypical structures. This analysis was done based on ERA5 data, because these three events occurred in the months from November to April, where we don't have OMPS-LP or S5P data (because of the polar night and OMPS-LP and S5P being satellite instruments that use the visible spectrum of the light). Therefore next (section 3.2), for the further throughout analysis of the connection between changes of tropopause height and ozone based on OMPS-LP and S5P data the time period of MOSAiC summer was chosen, where we have reliable OMPS-LP and S5P data. We proved the correlation between tropopause height changes for both the satellite and in-situ measurements. Finally (section 3.3), we aim to show that one can get dynamical information about cyclones out of OMPS-LP data and we defined the lowering of the 250 ppb ozone level as the key process in this connection. We also show that a cyclone identification with only OMPS-LP data is doable, which underlines that dynamical information about cyclones are present in the OMPS-LP data.

Yes, in addition and overall, our analysis verifies that the ERA5 data, so a gap-free, long term ozone dataset can be used in the future for a analysis to define more key processes of this connection. Out of this one then can look for in the OMPS-LP and S5P data. Still, the assimilation of satellite data for example in weather forecast models is a useful approach.

Those goals of our analysis had been made clearer throughout the paper.

3) I'm not clear as to why the analysis using the S5P data is done. Is this done to validate ERA5 ozone data? If so, why was there not more analysis done with ERA5? For example, how come you did not identify the cyclone borders with the ERA5 data as well? As you state, the reanalysis data has the benefit of being gap-free and covers a long time period.

Yes, the analysis using the S5P data is done to validate the ERA5 ozone data and to validate the connection between tropopause changes and total ozone column changes in the Arctic region. As stated in the previous answer the goal is not a cyclone identification with ozone data, but to prove the usefulness of satellite ozone data for an assimilation in weather forecast models for a dynamical information about synoptic events. Therefore we don't want to identify the cyclone borders with the ERA5 data, but only validate it for future analysis.

4) For Figure 4, is there a relationship between what you are showing here and the cyclones? Meaning you are showing the relationship between total column ozone and tropopause height, which makes sense, but how does this tie back into the results from Figures 1-3?

**In this case we were using the 4-PVU tropopause height as an indicator for the cyclones, which wasn't mentioned in the text. To better tie this back into the results from Figures 1-3 we added the surface pressure as an indicator for the cyclones in Figure 4 and implemented it in the discussion of the Figure 4.**

5)     Why do you show the order of Figures 13 and 14 the way you do? The time stamp for the cross section in Figure 14 is 2 hours earlier than Figure 13.

**The order of the Figures 13 and 14 was chosen in that way, because the Fig. 13 is the corresponding cross section of the OMPS-LP orbit shown in Fig. 12. We first introduce the cyclone that we investigate with the OMPS-LP ozone data and show that the ozone contour levels up to 15 km are influenced by the cyclone. We then selected a OMPS-LP cross section (Figure 14) that well illustrates that 250 ppb is a suitable ozone level for the analysis. Here the OMPS-LP orbit 44264 is a good example that illustrates the error-proneness of lower ozone levels like 80 ppb or 150 ppb.**

6)     Figures 1-3:

- The red curve indicating the 4-PVU tropopause height is missing.
- Add a bit more detail to the X-axis (i.e., in the text you reference very specific times that would be easier to locate in these figures with additional labeling on the X-axis).
- What do the vertical black dashed lines represent?

**We added the red curve indicating the 4-PVU tropopause height. The vertical black dashed lines mark specific cyclone events and the specific times we refer in the text to. This has been made clearer now by adding an explanation in the figure caption and adding more references to the black dashed lines in the text.**

7)     Sections 2 and 3 could be combined into a 'Data and Methods' section

**Sections 2 and 3 were combined.**

8)     Line 48: East coast of where?

**Added USA in the text.**

9)     Line 79: for a multidisciplinary -> for multidisciplinary

**Done.**

10)    Lines 155-156: 'Due to the smaller scale of the tropopause height', what does this mean?

The plotting scale of the tropopause height in Fig. 4 is much more detailed (3 km range) compared to the plotting scale of the tropopause height (which was missing in this version) in Figs. 1 – 3 (12 km range). The small lowering of the tropopause for Event 2 (~1km) is only really visible in Fig. 4. We changed the text to "*Fig. 4b shows better than Fig. 2, that also for Event 2, …*" to be more clearly.

11)    Lines 284-285: Which figure showed this?

We wanted to show this with Figure 4, but as mentioned in the answer to comment 4) we were using the 4-PVU tropopause height as an indicator for the cyclones without mentioning it in the text. With the correction of comment 4), Figure 4 now properly shows the connection between total ozone columns and cyclone events using the 4-PVU tropopause height mentioned in Lines 284-285.

---

## Author Response (AR2)

**Relations between cyclones and ozone changes in the Arctic using data from satellite instruments and the MOSAiC ship campaign**

**F. Monsees[1], M. Weber[1], A. Rozanov[1], J. P. Burrows[1], A. Rinke[2], R. Jaiser[2,3], and P. von der Gathen[2]**

[1]Institute of Environmental Physics (IUP), University of Bremen, Bremen, Germany
[2]Alfred Wegener Institute Potsdam, Helmholtz Centre for Polar and Marine Research, Potsdam, Germany
[3]Department of Life, Light and Matter, University of Rostock, Rostock, Germany

monsees@iup.physik.uni-bremen.de

**Author's Response**

**We thank the reviewer for the helpful feedback on our manuscript! The comment was implemented in the paper. Our response to the comment can be found below. The reviewer comment is given in red and our answer in black.**

**Referee #2**

Figure 10: The legend is a bit confusing. I am assuming that the red bar in the legend is just to indicate cyclones that occurred during the Polarstern expedition. But given that it is red, and red corresponds to a specific strength of cyclone, it could be taken that only cyclones with that color were those that occurred during the expedition. You could remove the legend altogether since the figure caption adequately describes what is in the figure.

**"Cyclones that impacted the Polarstern" was removed from the legend in Figure 10 for the stated reason. "Polarstern ozone sonde launches" was kept to make the figure for the most part self-explanatory.**